# Status and First Results from the KM3NeT neutrino telescope

**Evangelia Drakopoulou[1]\* on behalf of the KM3NeT Collaboration,**

**1** Institute of Nuclear and Particle Physics, N.C.S.R. "Demokritos", Patr. Gregoriou E and 27 Neapoleos Str., Agia Paraskevi, 15341, Greece

\* drakopoulou@inp.demokritos.gr

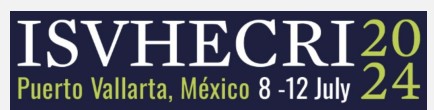

*22nd International Symposium on Very High Energy Cosmic Ray Interactions (ISVHECRI 2024) Puerto Vallarta, Mexico, 8-12 July 2024*

## Abstract

KM3NeT is a distributed research infrastructure under construction in abyssal sites of the Mediterranean Sea that hosts two underwater neutrino telescopes: ARCA, located offshore Portopalo di Capo Passero in Italy and ORCA, located offshore Toulon in France. Both telescopes employ the same photon detection technology but are opitmised according to different physics cases. ARCA is targeted to the detection of neutrinos with energies in the TeV-PeV range coming from astrophysical sources, while ORCA aims at studying the atmospheric neutrino oscillations at energies of a few GeV. In this contribution, the status of ARCA and ORCA is presented and the results obtained using data taken with the first detection units are discussed.

# 1 Introduction

The KM3NeT infrastructure consists of two neutrino telescopes, ARCA and ORCA, currently under construction and designed with different sizes and configurations. ARCA, deployed at a depth of 3500 m off the coast of Portopalo di Capo Passero, Sicily, Italy, is specifically optimised for detecting neutrinos from distant astrophysical sources in the TeV-PeV energy range. ORCA is situated at a depth of 2450 m off the coast of Toulon, France, and focuses on studying atmospheric neutrino oscillations at energies of a few GeV.

Both detectors leverage an innovative multi-PMT digital optical module (DOM) design [1], which significantly improves their detection capabilities. Each DOM is a pressure-resistant glass sphere that houses 31 three-inch photomultiplier tubes (PMTs), along with necessary calibration, positioning instruments [2], and readout electronics. These DOMs are used to measure Cherenkov light produced in seawater by charged particles resulting from neutrino interactions. By accurately recording photon arrival times and knowing the PMT positions, the trajectories of these particles are reconstructed. The spherical configuration of PMTs within each DOM ensures full $4\pi$ photon detection coverage. The DOMs are organised vertically along detection units (DUs), each DU comprising 18 DOMs secured by dyneema ropes [3]. ARCA and ORCA use different horizontal (90 m and 20 m, respectively) and vertical (36 m and 9 m, respectively) spacing between DOMs to target distinct energy ranges. Upon completion, ORCA will consist of 115 DUs, providing an instrumented mass of 7 Mton, while ARCA will have two 115-DU blocks, creating a cubic-kilometer water detection volume.

Currently, both KM3NeT detectors are operational and taking data with their initial DUs. Since December 2015, KM3NeT/ARCA has been deploying DUs and has already detected a number of neutrino candidates; at the time of writing it operates with 28 DUs (ARCA28), surpassing the ANTARES telescope [4] in capabilities and assuming its role in multi-messenger follow-up observations. Data collection at KM3NeT/ORCA also continues, with 23 DUs active and running at high uptime. The following sections will cover data obtained from early deployments, specifically the first 6 DUs for ORCA (ORCA6) and 6 to 21 DUs for ARCA (ARCA6-21).

# 2 Indirect Cosmic Rays Measurements in KM3NeT

The measurements of the flux of high-energy muons produced in cosmic ray (CR) air showers are important to assess the energy spectrum and the chemical composition of the primary CRs flux. The KM3NeT Collaboration has performed studies on the zenith distribution of the rate of high-energy muons arising from Extensive Air Showers (EAS) at depths of several kilometres with the initial configurations of both ARCA and ORCA neutrino telescopes [5] with 6 DUs. The minimal muon energy (at sea level) required to reach the KM3NeT detectors is approximately 500 GeV with the majority of muons having energies in the TeV range (thus originating mostly from primary CRs with energies around 10 TeV). Simulations using the latest QCD models have been compared with data from the ORCA6 (shown in Figure 1) and ARCA6 detectors. These comparisons showcase a deficit in the simulations with respect to the data at the approximately 40% level for atmospheric muons in the TeV-energy range. This deficit weakly depends on the muon inclinations or the muon energy at sea level and is compatible with the measurements and models of the TeV-scale muon flux [5]. A summary of the discrepancies in different muon

energies coming from different primary CRs energies is provided in [5], also indicating that the observed deficit of TeV muons could presage underestimations of the neutrino production in cosmic sources with respect to the flux of the accelerated nuclei and the gamma ray flux.

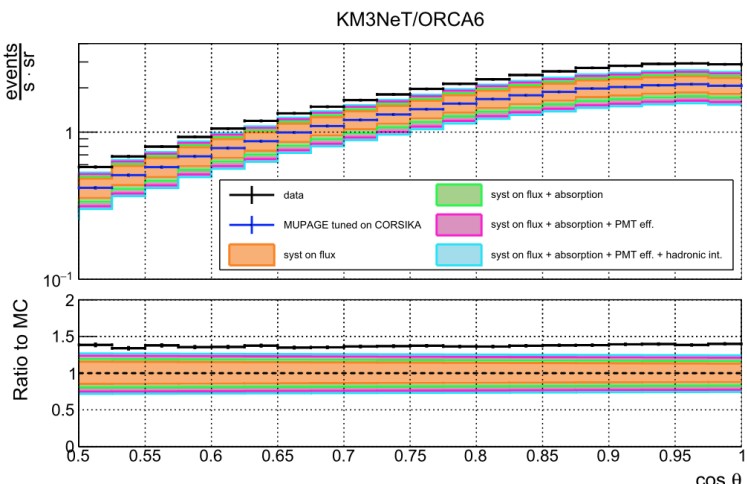

Figure 1: Top: The reconstructed muon rate as a function of the cosine of the zenith angle for ORCA6. The data points are shown in black, the simulations are in blue. Different systematic uncertainties are summed linearly and plotted as coloured bands. Bottom: The ratio between the data and the simulations. Statistical uncertainties are shown as vertical error bars [5].

The observation of the atmospheric muon shadowing effect due to the absorption of primary CRs by the Moon and the Sun can provide insights on the performance and the pointing accuracy of the KM3NeT detectors. To confirm the validity of the calibration and reconstruction procedures, the KM3NeT experiment has performed a first observation of the Moon and the Sun shadows in the sky distribution of cosmic-ray induced muons measured by the ORCA detector [6]. ORCA6 data, collected from February 2020 to November 2021, were used for this analysis. Despite the limited instrumented volume of the ORCA detector, at the early construction stage with 6 DUs, the shadows induced by the Moon and the Sun are detected at their nominal position, as shown in Figure 2. The statistical significance is 4.2 $\sigma$ and 6.2 $\sigma$, with an angular resolution of $\sigma_{res} = 0.49°$ and $\sigma_{res} = 0.66°$, respectively. These measurements are consistent with the prediction of 0.53° from simulations thus showcasing the good understanding of detector positioning, orientation and time calibration [7] and the accuracy of the event direction reconstruction [6].

# 3 The KM3NeT multi-messenger analysis

Multi-messenger astronomy is a rapidly advancing field that combines the information provided by different cosmic messengers: neutrinos, cosmic rays, gravitational waves and electromagnetic radiation to study transient astrophysical phenomena. Each messenger provides a distinct perspective on the Universe offering complementary information and enhancing the sensitivity to identify the sources of CRs.

To allow the coincident detection of astrophysical sources, systems for the distribution of external alerts and their follow-ups by multiple observatories have been developed worldwide. The KM3NeT Collaboration is currently developing an online platform to perform astronomy studies in real time for both ARCA and ORCA detectors and quickly alert the other telescopes if interesting events are observed. This real time analysis consists of two complementary pro-

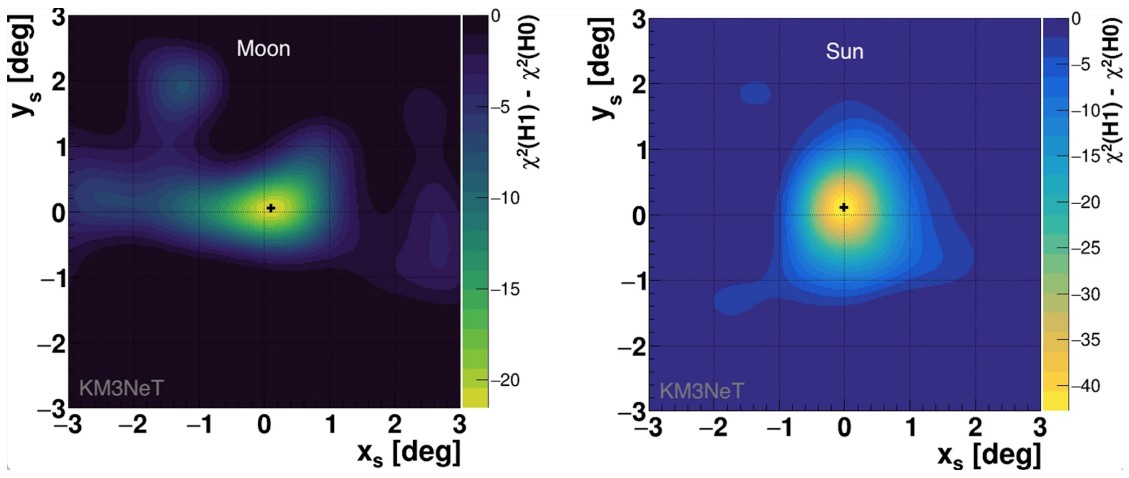

Figure 2: The Moon (left) and the Sun (right) CR shadow using ORCA6 data [6].

cedures: sending public alerts of potentially interesting events detected in KM3NeT and performing the follow-up of alerts detected by external observatories, to search for neutrinos in spatial and time coincidence [8]. The public alert sending is currently under development and is foreseen to be operational in early 2025. The follow-up activities have been ongoing since late 2022, conducting different analyses depending on the type of the alert and reporting results. No significant correlations have been found so far between the external alerts and the events detected with the partial ARCA and ORCA configurations. However, the sensitivity of observing cosmic neutrinos increases as the instrumented volume of KM3NeT expands.

## 4 ARCA: First results on Neutrino Astronomy

One of the main goals of the KM3NeT/ARCA is the detection of the diffuse flux of cosmic neutrinos aiming to provide information on the astrophysical sources and their acceleration mechanisms and on the production, composition and acceleration of CRs. The study of the diffuse neutrino flux also gives insights on signals from faint sources that are difficult to detect individually. The good direction reconstruction of the complete ARCA detector will allow for the identification of distant astrophysical neutrino sources. At neutrino energies above 100 TeV, where the background due to atmospheric neutrinos suppresses, the median angular resolution for the complete ARCA detector is at the order of $0.1°$ for track-like events (Charged Current $\nu_\mu/\overline{\nu}_\mu$, or Charged Current $\nu_\tau/\overline{\nu}_\tau$ with muon in the final state) and below $2°$ for shower-like events (electromagnetic and/or hadronic showers produced in CC $\nu_e/\overline{\nu}_e$, interactions, all other CC $\nu_\tau/\overline{\nu}_\tau$ interactions and all flavour NC $\nu/\overline{\nu}$ interactions).

The potential of the ARCA detector to measure a diffuse flux of astrophysical neutrinos with the partial configurations of ARCA with 6 to 21 DUs (ARCA6-21) is investigated under the assumption of an unbroken power law for the neutrino energy spectrum. In Figure 3 the convolution of sensitivities at 90% C.L. for ARCA6+8+19+21, as a function of energy, for a subset of selected spectral indices is shown. The small ARCA instrumented volume (representing approximately 18% of one building block of the complete detector), with the limited data taking of less than a year, is not yet competitive when compared to the ANTARES and IceCube experiments.

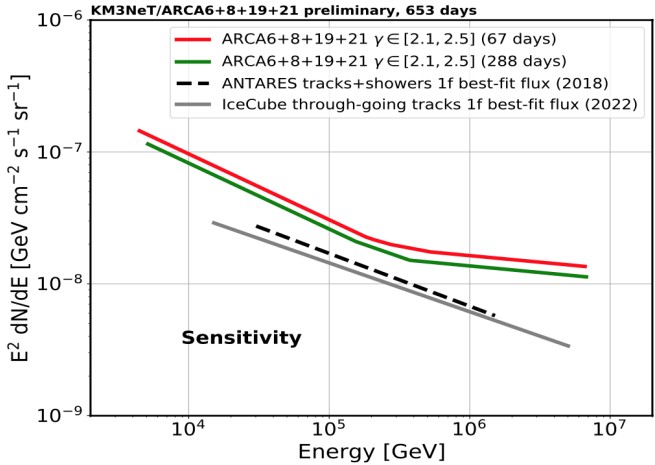

Figure 3: ARCA6-21 sensitivity to a diffuse flux of astrophysical neutrinos compared to the ANTARES and IceCube experiments [9] .

## 5  ORCA: First results on Atmospheric Neutrinos Studies

The main physics goal for KM3NeT/ORCA is the study of neutrino oscillations and neutrino mass hierarchy via the detection of atmospheric neutrinos. The most recent results on atmospheric oscillation parameters with the partial ORCA6 configuration are reported in [10]. The constraints on the oscillation parameters are in agreement with those obtained by other experiments and are expected to become more competitive as the ORCA instrumented volume is expanding. The inverted neutrino mass ordering hypothesis is disfavoured with a p-value of 0.25 [10].

Another important measurement achieved with ORCA6 data is the measurement of the atmospheric muon neutrino flux in the energy range between 1 GeV and 100 GeV, an energy region where only few measurements exist by other experiments. The energy spectrum of $\nu_\mu(\overline{\nu}_\mu)$ CC events is extracted by unfolding the experimentally measured energy distribution of the selected (with high-purity) atmospheric neutrinos events. Figure 4 shows the measurement which is in good agreement with the HKKM14 conventional flux model [12]. This measurement verifies the physics capabilities of the KM3NeT/ORCA detector even at the early construction stage in an energy region in which only few measurements exist.

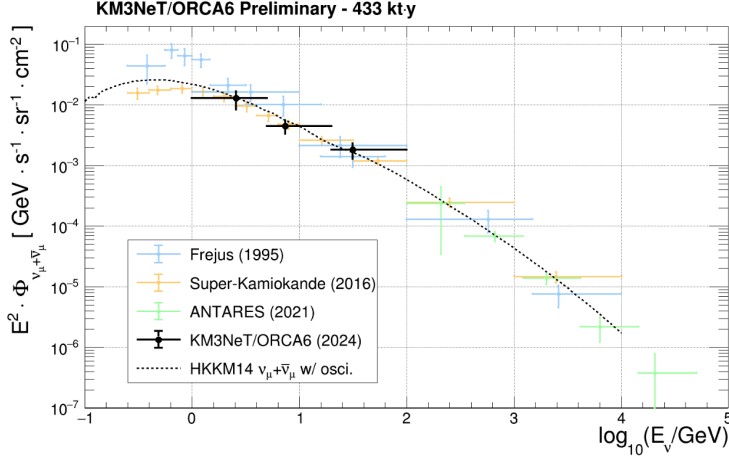

Figure 4: The atmospheric neutrino flux measurement using ORCA6 data is compared with measurements from ANTARES, Super-Kamiokande and Frejus [11].

## 6   Detection of an ultra-high energy cosmic neutrino

On the 13th of February 2023, KM3NeT/ARCA21 (ARCA with 21 DUs) recorded an ultra-high-energy event, named KM3-230213A [13]. The detected event - a muon produced by a neutrino interaction in the proximity of ARCA21 - has nearly horizontal trajectory and an estimated muon energy of approximately $120^{+110}_{-60}$ PeV. The extremely high energy of the muon (providing a lower bound on the incoming neutrino energy) and its horizontal directionality indicate an association with a cosmic neutrino. While the observed cosmic neutrino energy spectrum has shown a steep fall at high energies, the energy of this event significantly exceeds previous observations. This suggests that the neutrino originated from a different class of cosmic accelerators than those responsible for lower-energy neutrinos, or alternatively, that this may represent the first observation of a cosmogenic neutrino, produced when ultra-high-energy cosmic rays interact with relic photons in the Universe [13].

Figure 5 shows the position of KM3-230213A within the ($N_{PMTs}$, cos(zenith angle)) phase space. Figure 5(a) shows simulated Monte Carlo events, including the expected annual rates of atmospheric muons and cosmic neutrinos in ARCA21. Figure 5(b) shows the corresponding distribution from ARCA21 data, with KM3-230213A explicitly highlighted. The events are selected based on high-quality track reconstructions (well-reconstructed tracks) retaining 0.02% of all reconstructed atmospheric muon and neutrino tracks, and approximately 2% of cosmic tracks, assuming the flux model described in [13]. The detection of the first muon neutrino with an energy exceeding 100 PeV constitutes strong evidence for the presence of ultra-high-energy neutrinos in nature.

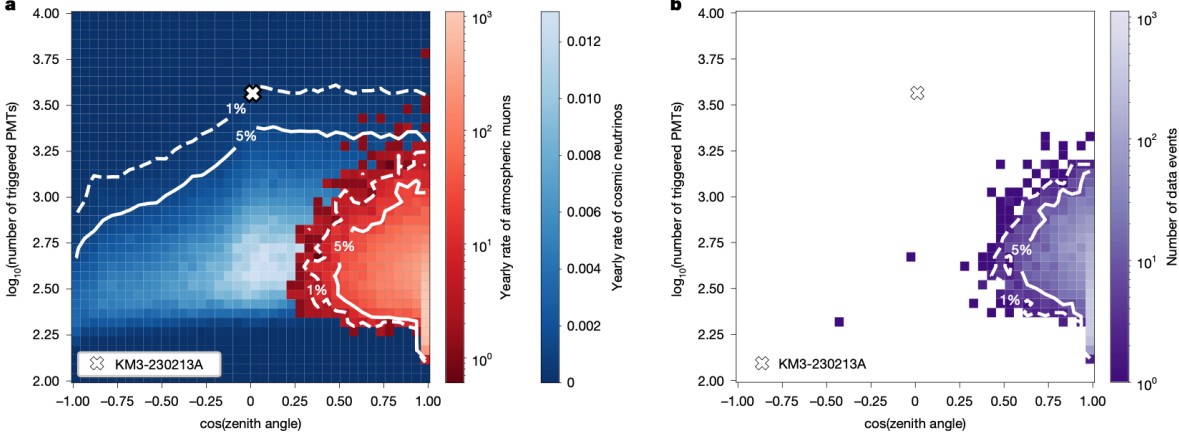

Figure 5: The expected yearly rate of atmospheric muons and cosmic neutrinos is shown for the ARCA21 detector as a function of the number of triggered PMTs and the cosine of the zenith angle. Solid and dashed contours correspond to regions containing 95% and 99% of the expected event distributions, respectively. The event KM3-230213A is indicated by a cross within this parameter space. Figure (b) shows the number of events recorded by ARCA21 over 287 days of data taking, with the same selection criteria. In addition to KM3-230213A, two lower-energy, upgoing neutrino candidates are also shown [13].

## 7   Conclusion

Although still under construction, the ARCA and ORCA detectors have shown good operation performances and provided the first interesting measurements, with their sensitivity rapidly

approaching that of long-established experiments. When completed, these neutrino telescopes
will allow for the full Sky coverage, including the Galactic Center, and detailed neutrino oscil-
lation studies while contributing to the new era of multi-messenger astronomy.

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
