# Peer review of "Status and First Results from the KM3NeT neutrino telescope"

_SciPost Physics Proceedings_

## Round 1 · Referee Report · Subir Sarkar (Referee 1) · 2025-3-20

Strengths

Reports results obtained using data taken with the first detection units of ARCA and ORCA, e.g.

  • Highlights deficit of ~40% in simulations of atmospheric muons in the TeV-energy range with respect to the data obtained with both ORCA6 and ARCA6

  • Confirms seeing Moon & Sun atmospheric muon shadow with expected resolutionusing ORCA6

  • Reports atmospheric neutrino flux measurements with ORCA6 in the 1-100 GeV range, in good agreement with standard Honda model as well as with pevious experiments

  • Shows expected sensitivity to astrophysical neutrinos of ARCA6, compared to previous experiments.

Weaknesses

Not a weakness as such - but this talk was given before the recent announcement of an extremely high energy cosmic neutrino by ARCA (Nature 638 (2025) 376), which is thus not mentioned.

Report

This was an invited submission from the KM3NeT collaboration to the 22nd International Symposium on Very High Energy Cosmic Ray Interactions. As such the speaker was nominated by the KM3NeT speaker's committee and the writeup of the talk was vetted by the KM3 NeT paper committee - as is standard practice for such a large astroparticle collaboration. Accordingly a `light touch' review ought to suffice in this case.

For example, it is reported that ORCA6 data (Feb 2020 to Nov 2021) on the Moon and Sun shadowing of cosmic ray produced atmospheric muons was used to validate the understanding of detector positioning, orientation and time calibration and the accuracy of event direction reconstruction. The Moon and Sun shadows were seen at 4.2σ and 6.2σ - agreeing with the prediction of 0.53 degrees from simulations. This has already been published in a refereed journal, accordingly it is neither necessary nor appropriate to ask for further details here.

This is a topical report from a running experiment; it is succint and well-written, so may be published.

Requested changes

If the author so wishes, they may add as a postscript the recent announcement of the detection of a muon with an estimated energy of 120 PeV - given the wide interest this has generated in the community.

Recommendation

Publish (surpasses expectations and criteria for this Journal; among top 10%)

---

## Round 2 · Author Response

Dear all,

Thank you for your suggestion! I have now added a section on the detection of an ultra-high energy cosmic neutrino by KM3NeT.

Kind Regards,

E. Drakopoulou

---

## Editorial Decision

accepted_in_target_journal